# Performance and Biomechanics in the Flight Period of Ski Jumping: Influence of Ski Attitude

**DOI:** 10.3390/biology11050671

**Published:** 2022-04-27

**Authors:** Lin Zhang, Xiong Li, Xin Wang, Long Chen, Tianyu Zhao

**Affiliations:** 1Department of Kinesiology, Shenyang Sport University, Shenyang 110102, China; zl_0202@126.com; 2Department of Mechanics, Northeastern University, Shenyang 110819, China; s4006861140@163.com (X.L.); chenlong@mail.neu.edu.cn (L.C.)

**Keywords:** sports performance, biomechanics, Computational Fluid Dynamics (CFD), V-style, tilted multi-vortex system

## Abstract

**Simple Summary:**

The adjustment of ski attitude during the flight period of ski jumping aims to improve the aerodynamic performance and thus enlarges the flying distance. Previous studies have measured the aerodynamic forces of an isolated ski through wind tunnel experiments; however, less information on the aerodynamic moment and underlying flow structures was provided. The biomechanic relation between the aerodynamics of the ski and the athlete’s ankle was also unknown. Using Computational Fluid Dynamics (CFD) methods, this research investigated the aerodynamic characteristics and related flow structures of a full-scale ski jumping ski in 125 attitudes. A convenient database for the aerodynamic forces and moments of the ski was established and the association between the aerodynamics of the ski and the control of the athlete’s ankle is discussed.

**Abstract:**

The performance of ski jumping is underpinned by multi-disciplinary principles, in which the aerodynamics of the ski dominates the flying distance and affects the biomechanics of the athletes’ ankle during the flight period. Conventional research on this topic was supported by wind tunnel experiments. Here, the aerodynamics of a full-scale ski jumping ski was calculated via Computational Fluid Dynamics (CFD) methods and good agreement with experimental data was achieved. The impacts of the angle of attack, yaw angle, and roll angle on the aerodynamic performance are explained. The inclusion of yaw angle can enhance the lift generation, which originates from the formation of a tilted multi-vortex system and the induced low-pressure footprints on the upper surface of the ski. Our results thus establish a database for the aerodynamic forces and moments of the ski and the associations between our findings and the skills in ankle control are discussed.

## 1. Introduction

Ski jumping is a winter sport that employs a pair of skis as the fundamental tool. A typical ski jump consists of four phases, i.e., in-run, take-off, flight, and landing, and, within most phases of ski jumping, the aerodynamic performance of the athletes, including their gear (skis, suits, goggles, and helmets) is of great importance [1,2,3,4,5], which resembles alpine skiing [6,7]. The score of ski jumping is averaged from the rating of five judges considering both the flying distance and style. Usually, the flying distance is related to the weight of the athletes, aerodynamic forces during flight, and the initial flight velocity after the take-off. Studies have revealed that the initial flight velocity is determined by the in-run approach, the take-off jumping velocity, and the posture transition during the take-off [8,9,10,11,12,13,14]. In contrast, the aerodynamic force during flight is dominated by the flying posture of the athletes and the skis. Due to the considerable number of degrees of freedom (DoFs) of the athlete’s body and the skis, determining the athlete-specific optimal posture during flight is a challenging task. Given the short take-off and flying periods, it is even harder for the athletes to make the decision during the flight.

One of the most important breakthroughs in the optimal flying posture was the change of flight style in the 1990s. Early ski jumping flight styles were all in H-style, i.e., the skis were in parallel (Figure 1a). Using computer simulations, Remizov developed an optimal control flight model for the H-style ski jumping, and the optimal time history of the angle of attack for maximizing flight distance was presented [15]. Later, the V-style was first used by Jan Boklov (Sweden) in 1985 and became one of the most successful flying postures of skis until now (Figure 1b). During the flight, the heads of the skis are spread out in the V-style, featured by a ski-opening angle [16]. Moreover, the athletes are allowed to lean further forward in the V-style, which improves the aerodynamic performance and leads to a longer flying distance. Mahnke and Hochmuth performed the first research on the V-style via a series of wind tunnel experiments [17]. Their results proved that the V-style athlete-ski system is superior to the conventional H-style positions considering the aerodynamic performance. Wind tunnel experiments were also conducted by Watanabe et al., who measured the aerodynamics of a full-scale athlete-ski model in both V-style and H-style [18], and Jin et al., who compared the performance of a scaled-down athlete-ski model in three jumping styles of ski positions, i.e., H-style, V-style and flat V-style [19]. The research of Jin et al. found that the flying distance of the model in the flat V-style is the longest (over 110 m), followed by the models in the V-style and classic H-style. If the ski positions can be changed during the flight, the best solution to maximize the flying distance (up to 112.5 m) is to switch from the H-style to the V-style at 1.3 s after the take-off, or from the flat V-style to the V-style at 1.6 s after the take-off.

Instead of comparing the V-style with the H-style, most research on the V-style position has discussed the effects of the ski-opening angle, combined with the variation of ski angle of attack (*α*). Using wind tunnel experiments, Seo et al. measured the aerodynamic lift, drag, and pitching moment of a V-style athlete-ski model during the free-flight phase, considering the variations of *α*, body-ski angle, and ski-opening angle [20]. An aerodynamic model for the ski was then established through a polynomial fitting of their measured data. To reduce the drag in the early flight phase and maximize the lift during the later flight phase, the ski-opening angle should be around 26°. Seo et al. further carried out a computer simulation of the flying distance based on the model and proposed an optimal control strategy for the ski-opening angle during flight [21]. Yoshida also performed a similar research procedure to the flight phase of ski jumping [22]. In addition, Nørstrud and Øye separated the aerodynamic forces of the athlete and ski using Computational Fluid Dynamics (CFD) tools (non-viscous Euler computations) [23]. They showed that the lift and drag of both the jumper and the skis peak when the ski-opening angle is between 20° and 30°. Moreover, flow visualization showed the shedding of the drag-producing vortex from the outer side of the ski tip, which motivated a novel ski design. Hu et al. also numerically investigated the influence of ski-opening angle on the aerodynamic force of skis [24]. The optimal ski-opening angle corresponding to the highest lift-to-drag ratio is also around 30° in their results. However, these experimental and numerical works on the V-style mostly ignored the coupling effects between the roll angle (*γ*), which limits their applications in the flight simulation of field jumping.

The *γ* of the skis inherently originates from the spreading of the athlete’s legs when achieving the ski-opening angle of the V-style. In the literature, the variation of *γ* in realistic field jumps and the contribution of *γ* to the aerodynamic performance of the skis are still underexplored. Cutter conducted wind tunnel measurements on the aerodynamics of a V-style ski jumping model considering the variation of the ski-opening angle and ankle angle [25]. In their setup, an increase in the ankle angle can reduce *γ*. The best lift-to-drag ratio (close to 1.55) was reported when the ski-opening angle and the ankle angle were 22.5° and 20°, respectively. As the ski rolls towards the horizontal plane, the lift-to-drag ratio can be improved, especially at a lower *α*. Moreover, Virmavirta and Kivekäs measured the aerodynamic forces of an isolated ski in the wind tunnel with various *α* (0–40°), ski-opening angle (0–40°), and *γ* (0–45°). They concluded that the *γ* corresponding to the maximum lift-to-drag ratio is around 5–10° when *α* is 30°, while no details on the flow structures were provided [26]. Despite this, these results encouraged the usage of curved sticks at the back part of the ski binding to maintain the horizontal position of the skis and thus improve the flight performance [26,27].

To further investigate the coupling effects between *α*, ski-opening angle, and *γ* on the aerodynamic features of ski jumping skis and uncover associated flow patterns, the CFD simulation of a single ski is conducted in this research. Although Meile et al. questioned the accuracy of the CFD method in calculating the aerodynamics of ski jumping [28], recent research showed that a well-established CFD model can mostly reflect the physics of ski jumping and the results become comparable to those from wind tunnel experiments [29,30,31,32,33,34,35]. The ski geometry and the numerical method are introduced in Section 2. The combined effects of three attitude angles (*α*, *β*, and *γ*, with *β* related to the ski-opening angle) on the aerodynamic performance and flow structures are reported in Section 3, followed by a discussion of our findings in Section 4. Finally, our concluding remarks are summarized in Section 5.

## 2. Materials and Methods

### 2.1. Geometry and Kinematics

The simplified geometric model of a ski jumping ski is characterized by a rectangular plate with a tilted semi-circle attached to the head (Figure 2a). The length (*l*), width (*b*), and thickness of the rectangular region are 2.42 m, 0.11 m, and 0.01 m, identical to a real ski jumping ski [26]. The up-tilting angle and radius of the semi-circle are 30° and 0.055 m. The reference area of the ski (*A*_ref_, including the semi-circle) is 0.2757 m^2^. The origin of the inertial frame (*x_i_y_i_z_i_*) is located at the center of the rectangular region, which is assumed to be the foot center of the athlete. As shown in Figure 2b, the attitude of the ski relative to the incoming airflow can be described by three successive rotating angles: pitch angle (*α*, same as the angle of attack in the literature), yaw angle (*β*, half of the ski-opening angle) and roll angle (*γ*). Note that the positive values of these attitude angles are defined in the clockwise direction in this research. The ranges of these attitude angles are between 0° and 40° to cover all possible ski positions in actual ski jumping. The velocity of the incoming airflow is fixed as 30 m/s, which lies in the range of realistic flying speed [26].

To simplify our simulations, the ski is maintained at its original attitude while the direction of airflow is varied to realize the prescribed *α*, *β*, and *γ* values. The rotation of airflow relative to the ski is achieved by three rotation matrices,
(1)$$v^{\prime }=R\left(\gamma \right)R\left(\beta \right)R\left(\alpha \right)\left[\begin{array}{c} U_{\infty }\\ 0\\ 0 \end{array}\right]$$
with
(2)$$R\left(\alpha \right)=\left[\begin{array}{ccc} \cos\alpha & -\sin\alpha & 0\\ \sin\alpha & \cos\alpha & 0\\ 0 & 0 & 1 \end{array}\right],$$
(3)$$R\left(\beta \right)=\left[\begin{array}{lll} \cos\beta & 0 & \sin\beta \\ 0 & 1 & 0\\ -\sin\beta & 0 & \cos\beta \end{array}\right],$$
and
(4)$$R\left(\gamma \right)=\left[\begin{array}{ccc} 1 & 0 & 0\\ 0 & \cos\gamma & -\sin\gamma \\ 0 & \sin\gamma & \cos\gamma \end{array}\right].$$

Here, *U*_∞_ denotes the magnitude of the incoming airflow. The Reynolds number (Re) is then calculated as,
(5)Re=U∞lυ=4.8×106
where υ = 1.5 × 10^−5^ m^2^/s is the kinematic viscosity of the airflow.

### 2.2. Numerical Setup

#### 2.2.1. Meshing

The fluid domain containing the ski is outlined by a sphere (radius = 30 m) around the center of the inertial frame (Figure 3). Moreover, another concentric sphere region (radius = 4.25 m) with refined meshes is defined around the ski. All surface meshes are hexagons and the volume mesh is generated by the polyhedral method with hex-cores. This ensures that most of the volume meshes are hexahedrons, which is beneficial to the convergence of iteration. Two buffer layers are inserted between the polyhedron cells and hex-cores for the transition. The boundary layer above the surface is meshed by 12 layers of hexahedrons, with the first grid distance at 4 × 10^−5^ m (the corresponding *y*^+^ at the steady-state is limited to around 1). The size of the computational mesh is thus determined by the maximum size of the surface meshes. The typical mesh used in this research possesses 1,300,000 cells, with the maximum sizes of surface meshes being 0.002 m and 3 m for the ski and far-field, respectively.

#### 2.2.2. Numerical Scheme and Boundary Conditions

The governing equations for the flow over the ski are the incompressible 3-D Navier-Stokes equations,
(6)∇⋅v=0
and
(7)$$\frac{\partial v}{\partial t}+\left(v\cdot \nabla \right)v=f-\frac{1}{\rho }\nabla p+\upsilon \nabla^{2}v$$

Here, **v**, *ρ*, and *p* are the velocity vector, air density, and stative pressure, respectively. **f** denotes the external force acting on the fluids, i.e., the gravity in this research. The governing equations were solved by a finite-volume-based implicit RANS (Reynolds-Averaged Navier-Stokes) solver. The momentum equation was solved using a second-order upwind scheme and the temporal discretization was achieved by the first-order implicit formulation. Following previous CFD studies on ski jumping [31,33], the *k*-*ω* SST (shear stress transient) model was employed in the solver to account for the flow transition from laminar structures to turbulent structures. The *k*-*ω* SST model combines the advantages of conventional *k*-*ε* and *k*-*ω* models [34] and Defreaeye demonstrated that the *k*-*ω* SST turbulence model shows good agreement with experimental data for cyclist aerodynamics [35].

The boundary conditions of our numerical solution are summarized as follows. At the far-field of the fluid domain, a uniform constant incoming airflow **v**′ with a magnitude of 30 m/s is imposed. The direction of **v**′ can be calculated by Equations (1)–(4). The ski surface is imposed by the boundary condition of non-slip walls.

#### 2.2.3. Data Analysis and Validation

The aerodynamic forces and moments of the ski are calculated by an integral of pressure stress (normal to the surface) and friction stress (parallel to the surface) acting on each surface cell. The center of the moment is defined at the origin of the inertial frame. To better relate the aerodynamic forces and moments of the ski to the athletes, the forces and moments vectors are projected into the airflow frame (*x_γ_y_γ_z_γ_*), i.e., lift (*L*), drag (*D*), side force (*S*), pitch moment (*M_z_*), yaw moment (*M_y_*), and roll moment (*M_x_*). Due to the relative rotation of the airflow w.r.t. the ski, the unit vector (**e***_γ_*) indicating the directions of the airflow frame can be calculated by (**e***_i_* represents the corresponding unit vector in the inertial frame):(8)$$e_{\gamma }=R\left(\gamma \right)R\left(\beta \right)R\left(\alpha \right)e_{i}$$

The dimensionless coefficient corresponding to the aerodynamic forces and moments are defined as follows,
(9)$$C_{L}=\frac{L}{0.5\rho U_{ref}^{2}A_{\mathrm{Ref}}},\;C_{D}=\frac{D}{0.5\rho U_{ref}^{2}A_{\mathrm{Ref}}}$$
and
(10)$$C_{M,i}=\frac{M_{i}}{0.5\rho U_{ref}^{2}A_{\mathrm{Ref}}l}$$

Here, *M_i_* denotes the aerodynamic moment and the subscription *i* represents pitch, yaw, and roll components. The lift-to-drag ratio (*L*/*D*) is thus equivalent to the ratio of their coefficients *C_L_*/*C_D_*.

To guarantee a converged solution, the mesh size of our numerical setup was validated via a comparison with previous wind tunnel measurements [26]. A typical ski position with *α* = 30°, *β* = 15°, and *γ* = 0° is selected and the mesh size is varied by changing the size of surface meshes. As shown in Figure 4a, the *C_L_* and *C_D_* almost converge as the mesh reaches 1,000,000 cells (1 M). Further refinement of the mesh only results in trivial fluctuations while the computational cost is significantly enlarged. Therefore, the mesh with 1,300,000 cells was used in this research. We further compared our prediction on the *C_L_* and *C_D_* of a ski with *α* = 30°, *β* = 0°, and *γ* = 0° to those measured from the wind tunnel experiments. As shown in Figure 4b, the errors in the *C_L_* and *C_D_* are 2.6% and 13.3%, respectively. These errors may be attributed to the simplification in the ski model and the roughness of the ski surface, which were not provided in the experiments. Nevertheless, the comparison proves that our numerical method can present a reasonable estimation of aerodynamic forces and moments of ski jumping skis.

## 3. Results

### 3.1. Aerodynamic Forces and Moments

The aerodynamic forces and moments of the ski at 125 positions are shown in Figure 5 (the detailed database is listed in the Appendix A). In general, the lift and drag of the ski are enlarged as the angle of attack (*α*) increases toward 30°. A further increase of *α* from 30° to 40 can further enhance the drag while the lift becomes saturated. At *α* < 20°, the increase of *β* and *γ* from 0° to 40° can both continuously enhance the lift and drag of the ski, while the high lift-to-drag ratios are obtained when almost no *β* and *γ* are included. This infers that the drag increment is more sensitive to the yaw and roll of the ski at a low *α*. At *α* > 20°, the combination of a large *β* and a small *γ* is favored by the ski to maintain a high lift and drag. Specifically, the lift maximum is achieved around *α* = 30°, *β* = 20°, and *γ* = 0°. This suggests that the lift enhancement led by the yaw motion is valid within a limit of *β* at an *α* close to the stall. At high *α* and *β* combinations, the inclusion of roll motion is consistently detrimental to the lift enhancement. The drag experiences a similar impact of *β* and *γ* at a higher *α*, except that the increases of *γ* at a high *β* can result in a further drag increment. Considering the lift-to-drag ratio (*L*/*D*), the increase of *α* is consistently adverse to a high *L*/*D* value and the impact of *β* and *γ* on *L*/*D* becomes trivial as *α* goes up. Thus, it is indicated that the inclusion of yaw and roll motion can improve the lifting capacity of the ski while almost retaining the *L*/*D*, which is beneficial to achieving a longer flying distance.

The aerodynamic moments corresponding to the airflow (*C_M_z_*, *C_M_x_*, and *C_M_y_*) are shown in Figure 5d–f. Within the range of *α*, *β*, and *γ*, *C_M_z_* experiences a more significant variation than *C_M_x_* and *C_M_y_*. In general, the increase of *α* consistently enlarges the *C_M_z_* and the inclusion of *β* can lead to a further increment. Note that the *C_M_z_* in our parameter space is negative, corresponding to a nose-up pitching moment with respect to the athlete. The impact of *γ* is remarkable at a moderate *α* (*α* > 20°) when *C_M_z_* is reduced as *γ* goes up. Although a stall phenomenon is observed in the lift around *α* = 40°, no decrease or saturation of *C_M_z_* occurs as *α* reaches 40°. In contrast, a consistent enlargement of *C_M_x_* is observed as both *β* and *γ* increase from 0° to 40° at all *α* (Figure 5e). The *C_M_x_* is mostly negative (clockwise) in the research, indicating pronation around the ankle. In addition, *C_M_y_* is barely changed until *α* > 20°, when an increase of *γ* results in a reduction (from a positive *C_M_y_* to a negative *C_M_y_*). A negative *C_M_y_* tends to further extend the ski-opening angle and thus imposes a stronger reacting moment to the athlete’s foot. Compared to *C_M_z_* and *C_M_x_*, the magnitude of *C_M_y_* is less comparable at almost all *α*.

### 3.2. Flow Structures and Pressure Analysis

#### 3.2.1. Effects of α

The flow structure and pressure distribution of the ski at various *α* are shown in Figure 6. The *β* and *γ* are 20° and 0°, respectively, corresponding to the lift peak at a high *α*. Theoretically, according to the aerodynamics of a slender body at high *α* but neutral *β* and *γ*, two vortices should be generated along both side edges of the ski. However, the inclusion of *β* breaks down the symmetry of the flow structure, which transforms into a multi-vortex system. At *α* = 10° (Figure 6a), a triple-vortex system (V1 to V3) is produced above the upper surface and the vortex system is tilted towards the right side of the ski due to the positive *β*. The low-pressure region (LPR) on the dorsal surface of the ski, which forms a longitudinal strip, is mostly induced by V1 and V2. Moreover, no significant high-pressure region (HPR) is produced on the upwind surface. Thus, limited pressure force is formed normal to the ski. Since the pressure force is the major source of lift and drag on the ski, the limited pressure force at a lower *α* leads to the lower lift and drag.

As *α* increases to 20°, additional vortices are generated from both side edges and a robust Hexa-vortex system (V1 to V6) is formed. The intervals between vortices along both side edges are reduced and all vortices are tilted towards the right side of the ski. Therefore, each vortex from the left side covers a slender LPR region on the upper surface. Moreover, the vortices from the right side can induce a local LPR around their attachment points. As a result, the strip-like LPR region at the lower *α* is transformed into a footprint pattern, which becomes remarkable at *α* = 30°. These stronger LPR footprints at *α* = 30° are attributed to the enhanced vortex intensity and the approaching of the Hexa-vortex system towards the ski. Together with the stronger HPR on the upwind surface, the pressure force is significantly enhanced at *α* = 30°.

When *α* reaches 40°, the LPR footprints are attenuated and almost no remarkable footprints are formed around the tail of the ski. Despite the HPR on the upwind surface being barely changed, the pressure force is reduced at *α* = 40°, corresponding to the stall phenomenon in lift generation. The attenuated LPR around the tail is led by the tilted-up V4–V6, the low-pressure regions within which are convected away from the ski. However, the lift reduction at *α* = 40° is mostly located behind the mid-point of the ski, indicating that the nose-up pitching moment (*C_M_z_*) around the athlete’s foot is further enlarged (as shown in Figure 5d).

#### 3.2.2. Effects of β

The effects of *β* on the flow structure and pressure distribution of the ski are shown in Figure 7. Note that the comparative case with *α* = 30°, *β* = 20°, and *γ* = 0° is shown in Figure 6c. According to Figure 6c, the ski with neutral yaw and roll angles should experience a symmetrical flow pattern and pressure field. The LPR on the dorsal surface is limited around the head of the ski, and no LPR footprints are observed. This is because no tilted multi-vortex system is generated and the symmetrical double-vortex system at *β* = 0° and *γ* = 0° is lifted along the ski, reducing the induced LPR. However, when the yaw angle is further enlarged (*β* = 40°), the LPR footprints are also attenuated and shrink towards the head of the ski, which is similar to an increase of *α.* The absence of LRP footprints at *β* = 40° can be separated into two aspects. First, the low-pressure regions within the vortices of the Hexa-vortex system are weaker. Second, the vortices from the left side are tilted further away from the ski, limiting the influence of these vortices on the upper surface. As a result, a nominal stall region is observed behind the mid-point of the ski. Consequently, the lift and drag increase at an asymmetric yaw position is explained by the formation of a tilted multi-vortex system and LPR footprints. Nevertheless, the improvement is only valid within a certain limit of yaw angle, beyond which a stall can be motivated.

#### 3.2.3. Effects of γ

Figure 8 illustrates the flow structures and pressure distributions of the ski at different roll positions. As shown in Figure 6c, the typical flow structure around the ski at *α* = 30°, *β* = 20°, and *γ* = 20° is characterized by a tilted Hexa-vortex system, which induced the LPR footprints on the upper surface. As *γ* increases to 20°, the roll motion of the ski indicates a pronation control, and the flow structure of the ski is downgraded into a triple-vortex system, which resembles that at a lower *α* (Figure 6a). However, the intensity of the triple-vortex system at a higher *γ* is stronger than that at a lower *α*, thus the strip-like LPR is stronger. As *γ* further increases to 40°, the triple-vortex system and the corresponding strip-like LPR are both reversed with respect to the longitudinal axis of the ski (Figure 8b). The HPR on the upwind surface also experiences an identical variation as *γ* goes up. Therefore, the pressure force and the resulting lift and drag are barely changed at *γ* > 20°. These findings indicate that the detrimental impact of *γ* on the aerodynamic performance of the ski is remarkable when *γ* starts to deviate from zero and the contributions are equivalent to a decrease of *α*.

## 4. Discussion

In Section 3, considerable CFD simulations were conducted to obtain the aerodynamic features of a ski jumping ski, which establishes a database for the aerodynamic forces and moments. In this section, we will further discuss the associations between the database and its guidance for the skills and performance of ski jumping.

Let us now discuss the trade-off between the peaks of lift and lift-to-drag ratio (*L*/*D*). As shown in Figure 5c, the *L*/*D* peak is achieved at a neutral angle of attack (*α* = 0°), when the ski is confined within the plane of incoming airflow. For any combination of yaw and roll angles (*β* and *γ*) in this plane, the drag (*D*) mostly originates from the friction stress (tangential to the ski) instead of the pressure stress (normal to the ski), which becomes dominant at a positive *α*. Thus, *D* is significantly lower at a neutral *α*, leading to a boost in the *L*/*D*. However, the lift coefficient (*C_L_*), is also below 0.3 due to the limited pressure force. This is, without a doubt, not encouraged to be used in the field jump, since the balancing effect of the lift (with respect to the gravity) in the vertical direction is trivial and the ski almost undertakes a free-fall with the initial jumping velocity. According to previous computational simulations of ski jumping [21,32,33], a trade-off between the lift and the lift-to-drag ratio is inferred. In other words, the optimization of the lift-to-drag ratio should be conducted with a constraint of lift minimum. For example, if the athlete places the ski at *α* = 30°, the *C_L_* maximum is achieved when yaw and roll angles are around 20° and 0°, where the *L*/*D* is also close to its peak (at this specific α). This critical attitude resembles that reported by Virmavirta and Kivekäs when the optima yaw and roll angles are 15° and 5° at *α* = 30° [26]. The training of flight posture can focus on this critical ski attitude to prolong the flying period.

The flow structures at *β* > 0° demonstrate the underlying physics of lift enhancement due to the yaw angle, i.e., the footprints of the low-pressure region. Since these footprints are induced by the roll-up of a multi-vortex system at both side edges of the ski, it is reasonable to reshape the local side edge at the attaching points of those vortices to either intensify the vortices or attract the vortices towards the upper surface. This will also enlarge the footprints and thus the pressure force. Note that the specific modification of the side edge is strongly relevant to the ski attitude. Our research suggests that this modification can start with the optimum attitude discussed above.

Moreover, the increase of *γ* can act as a reduction of *α*, which is most detrimental to flying performance. However, as the athlete’s legs are spread out to a target yaw angle, an increase of *γ* is inevitably involved. Therefore, our research encourages the athletes to pronate their feet to compensate for the increase of *γ* and improve the aerodynamic performance of the ski. This also supports the usage of a curved stick at the back part of ski bindings to assist in balancing the increase of *γ*.

Finally, we must declare the limitations of the current study. First, our CFD simulations are based on a single ski jumping ski and ignore the interactions between two skis, as well as the interactions between the skis and the athlete. For example, due to the symmetry, the yaw and roll moments generated by a pair of skis should be canceled and their overall impacts on the athlete’s body should be trivial. However, these moments should be considered when analyzing the equilibrium of each foot. Second, although most details of the ski geometry are considered in the simplified model, the surface roughness, which determines the friction stress is not considered. Third, the yaw angle defined in our research may not be equal to half the magnitude of the ski-opening angle in some research. Since the ski-opening angle (or V angle) is not explicitly defined in most previous research, we have to set up a new frame system to consistently introduce the yaw angle. Any potential application of our data in the future should notice this difference.

## 5. Conclusions

Our results establish a convenient database for the aerodynamic forces and moments of the ski under various combinations of the angle of attack, yaw angle, and roll angle. An optimal ski attitude (angle of attack, yaw angle, and roll angle equal to 30°, 20°, and 0°, respectively) to prolong the flying period is suggested. The connections between our findings and the skill training in ski jumping are discussed and advice for ankle control is suggested. The lift enhancement due to the yaw angle originates from the formation of a tilted multi-vortex system and the induced low-pressure footprints on the upper surface of the ski. The increase of roll angle is detrimental to both lift generation and lift-to-drag ratio, and therefore foot pronation is encouraged for the athletes when spreading their legs during the flight. Further analysis shows that an excessive increase in yaw angle up to 40° can also motivate a nominal stall, which indicates that the benefits from the yaw angle are only valid within a certain range.

## Figures and Tables

**Figure 1 biology-11-00671-f001:**
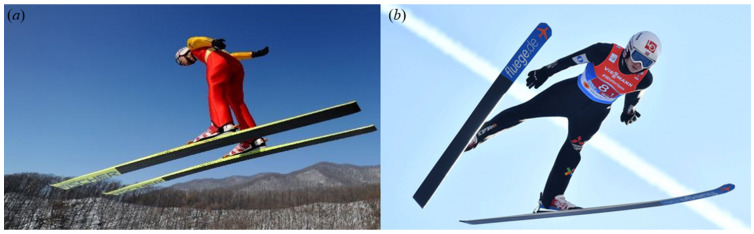
Two typical flight styles during the flight of ski jumping: (**a**) H-style (athlete: Guang Yang, photo: Men’s K90.3 m Ski Jumping Team Competition in the 12th National Winter Games). (**b**) V-style (athlete: Halvor Egner GRANERUD, photo: Ski Jumping World Cup).

**Figure 2 biology-11-00671-f002:**
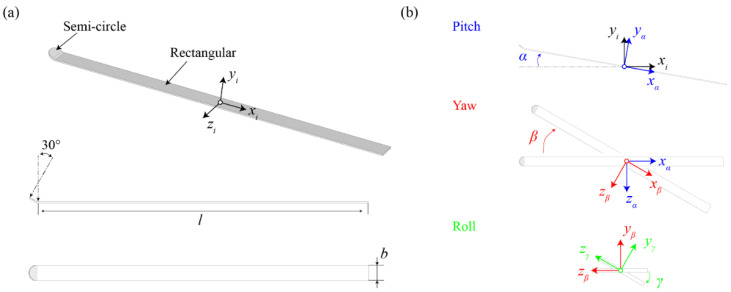
Schematics of a ski jumping ski: (**a**) geometric design and (**b**) definition of attitude angles.

**Figure 3 biology-11-00671-f003:**
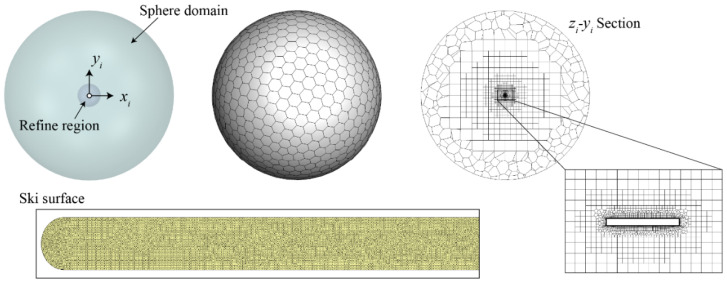
The computational mesh of the fluid domain and the ski.

**Figure 4 biology-11-00671-f004:**
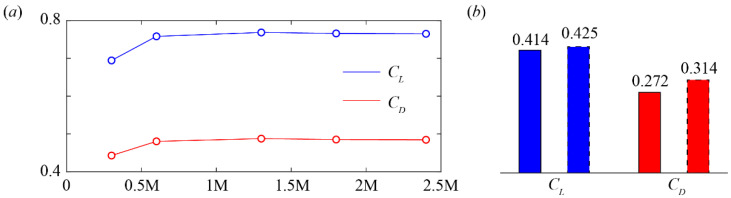
Validation of the numerical method. Unit M in (**a**) denotes a million and the experimental data in (**b**) is outlined by dotted lines.

**Figure 5 biology-11-00671-f005:**
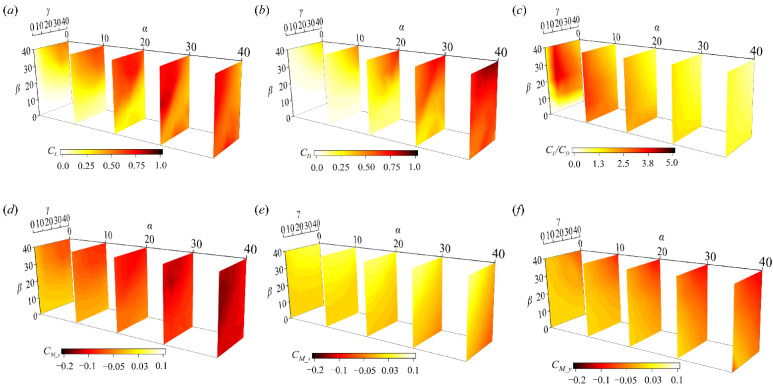
Aerodynamic forces and moments of the ski at different attitude angles: (**a**) lift, (**b**) drag, (**c**) lift-to-drag ratio, (**d**) pitch moment, (**e**) yaw moment, and (**f**) roll moment.

**Figure 6 biology-11-00671-f006:**
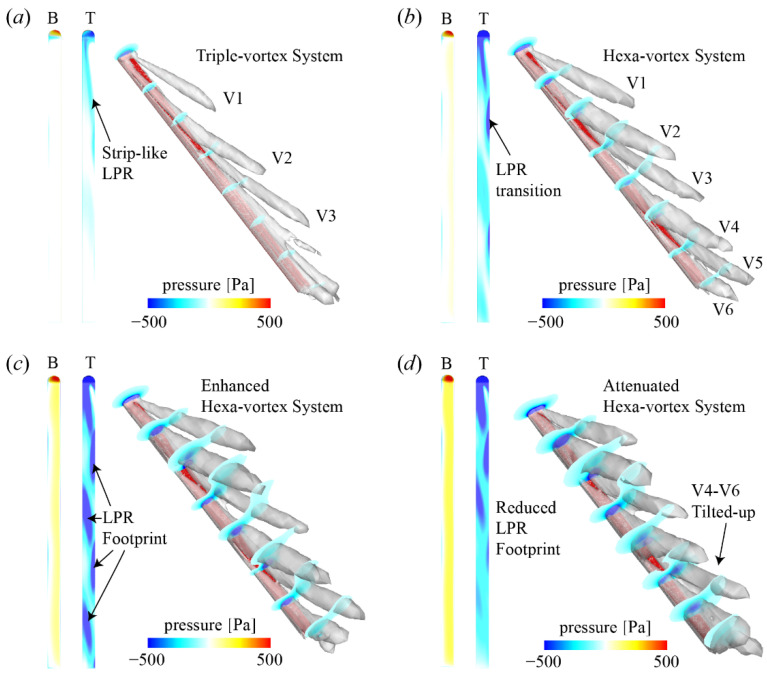
Flow structure and pressure distribution of the ski at *β* = 20° and *γ* = 0°: (**a**) *α* = 10°, (**b**) *α* = 20°, (**c**) *α* = 30°, and (**d**) *α* = 40°. B: bottom view; T: top view; V: vortex. The ski is labeled by the red region.

**Figure 7 biology-11-00671-f007:**
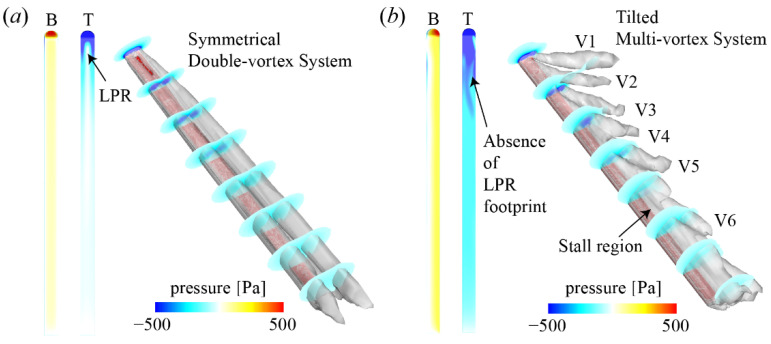
Effects of *β* on the flow structure and pressure distribution of the ski: (**a**) *α* = 30°, *β* = 0°, and *γ* = 0° and (**b**) *α* = 30°, *β* = 40°, and *γ* = 0°. B: bottom view; T: top view; V: vortex. The ski is labeled by the red region.

**Figure 8 biology-11-00671-f008:**
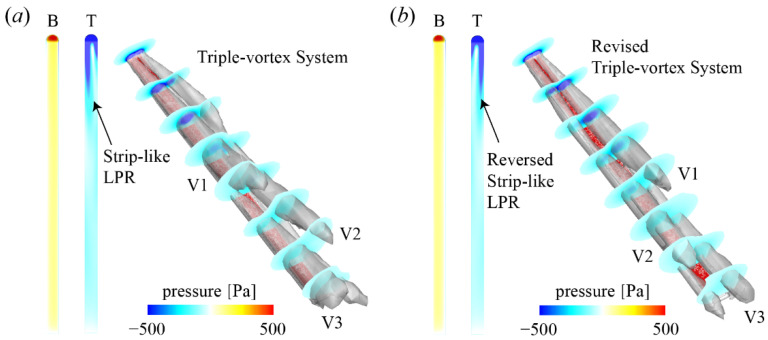
Effects of *γ* on the flow structure and pressure distribution of the ski: (**a**) *α* = 30°, *β* = 20°, and *γ* = 20° and (**b**) *α* = 30°, *β* = 20°, and *γ* = 40°. B: bottom view; T: top view; V: vortex. The ski is labeled by the red region.

## Data Availability

The data presented in this study are available on reasonable request from the corresponding author.

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
