# Peer review of "Performance and Biomechanics in the Flight Period of Ski Jumping: Influence of Ski Attitude"

_biology, 2022, doi:10.3390/biology11050671_

Round 1

Reviewer 1 Report

I have just read this paper with high intrigue. It seems normal to think about how to improve performance, but in this case, I was surprised by the line of research on the improvement of jump in ski sport. I reckon this work could have improved knowledge that we have about how to reduce drag and get a better performance in this sport. For that reason, I commend the authors for their effort.

Nevertheless, I do have few minor concerns for the authors to review. I hope the authors will find these suggestions helpful and they could improve the great manuscript done.

Keywords

  • I suggest to change “ski jumping” due to it is in the title, you can use another word so as to extend your visibility.

Introduction

  • Line 63. Correct the number use to cite the reference.
  • From line 81 to 83. Why you are writing about Seo's research if you are using Mahnke's research in the references?
  • From line 86 to 88. Why you are writing about Seo's research if you are using Watanabe's research in the references?
  • Line 95. The reference of Hu et al. is wrong in the cite or in the references. Revise it.
  • From line 108 to 112. Why you are writing about Virmavirta's research if you are using Qi's research in the references?
  • Line 117. Where is Meile’s reference?
  • Line 118. The reference use “26” is not the correct one. Revise it.

Results

  • Line 369. The reference use “23” is not the correct one. Revise it.

Conclusion

  • Perhaps it is too general.

References

  • Revise reference part due to there are different reference formats and maybe the order is not the correct one.

Author Response

 Response to Reviewer

We wish to thank the reviewer for providing detailed and insightful comments, and we hope all issues have been addressed satisfactorily.

  • Original comments are colored blue.
  • A point-to-point response can be found below.
  • All major changes in the manuscript are highlighted.

I have just read this paper with high intrigue. It seems normal to think about how to improve performance, but in this case, I was surprised by the line of research on the improvement of jump in ski sport. I reckon this work could have improved knowledge that we have about how to reduce drag and get a better performance in this sport. For that reason, I commend the authors for their effort.

Nevertheless, I do have few minor concerns for the authors to review. I hope the authors will find these suggestions helpful and they could improve the great manuscript done.

Keywords

  1. I suggest to change “ski jumping” due to it is in the title, you can use another word so as to extend your visibility.

Reply 1)

The keyword “ski jumping” has been replaced by “tilted multi-vortex system”.

Introduction

  1. Line 63. Correct the number use to cite the reference.

Reply 2)

We are sorry for the mistakes in the references and their corresponding citations in the main text. Following corrections w.r.t. the referencing have been all taken into consideration. The whole list of references, as well as the citations in the main text, have been carefully revised.

  1. From line 81 to 83. Why you are writing about Seo's research if you are using Mahnke's research in the references?
  2. From line 86 to 88. Why you are writing about Seo's research if you are using Watanabe's research in the references?
  3. Line 95. The reference of Hu et al. is wrong in the cite or in the references. Revise it.
  4. From line 108 to 112. Why you are writing about Virmavirta's research if you are using Qi's research in the references?
  5. Line 117. Where is Meile’s reference?
  6. Line 118. The reference use “26” is not the correct one. Revise it.
  7. Line 369. The reference use “23” is not the correct one. Revise it.

Conclusion

  1. Perhaps it is too general.

Reply 10)

Thanks for the suggestions. The conclusion has been revised to further detail our key findings.

References

  1. Revise reference part due to there are different reference formats and maybe the order is not the correct one.

Reply 11)

The reference part has been revised to avoid any disorder and format mistakes.

Reviewer 2 Report

The manuscript entitled "Performance and Biomechanics in the Flight Period of Ski Jumping: Influence of Ski Attitude" reports the results of a numerical study of aerodynamic phenomena during the flight phase of a ski jump. The authors focus on the air flow around the skis and thus find the air pressure distribution on both surfaces of the skis. In contrast to the conventional approach, which involves the use of a wind tunnel, the calculations are performed here using Computational Fluid Dynamics methods and a good agreement with experimental data is claimed. The attitude of the ski is discribed by three angles that determine its pitch, yaw, and roll. The important novelty of the approach reported in the manuscript is inclusion of the coupling between the yaw angle, which corresponds to the width of the V-style position of the athlete, and the roll angle of the skis. This results in the formation of a tilted multi-vortex system which may improve the generation of the lift force.

Specific comments.

Line 38, please consider removing "famous"

Line 39, please consider changing "jumping" to "jump"

Line 48, please change "degrees" to "number of degrees"

Line 49, please consider changing "finding out" to "determining"

Line 49, please consider changing "athlete-specified" to "athlete-specific"

Line 58, please replace "becomes" with "became"

Line 63, [16] instead of [1616]?

Line 135, please check the value of the Aref, the one in the text seems to be a little to large given the size of the ski defined a few lines earlier

Line 157, please give unit of the kinematic viscosity

Lines 210-214, please provide a more detailed explanation of how CL and CD are related to Eqs. (9) and (10)

Lines 434-443, are the references [6], [7], and [8] appropriate? Do they refer to ski jumping?

Lines 459-460, Forschungsinstitut

Figures 6, 7, 8, pressure and its unit, p/Pa, would be better reported as pressure [Pa] or pressure (Pa)

Author Response

 Response to Reviewer

We wish to thank the reviewer for providing detailed and insightful comments, and we hope all issues have been addressed satisfactorily.

  • Original comments are colored blue.
  • A point-to-point response can be found below.
  • All major changes in the manuscript are highlighted.

The manuscript entitled "Performance and Biomechanics in the Flight Period of Ski Jumping: Influence of Ski Attitude" reports the results of a numerical study of aerodynamic phenomena during the flight phase of a ski jump. The authors focus on the air flow around the skis and thus find the air pressure distribution on both surfaces of the skis. In contrast to the conventional approach, which involves the use of a wind tunnel, the calculations are performed here using Computational Fluid Dynamics methods and a good agreement with experimental data is claimed. The attitude of the ski is described by three angles that determine its pitch, yaw, and roll. The important novelty of the approach reported in the manuscript is inclusion of the coupling between the yaw angle, which corresponds to the width of the V-style position of the athlete, and the roll angle of the skis. This results in the formation of a tilted multi-vortex system which may improve the generation of the lift force.

Specific comments.

  1. Line 38, please consider removing "famous"

Reply 1)

Revised.

  1. Line 39, please consider changing "jumping" to "jump"

Reply 2)

Revised.

  1. Line 48, please change "degrees" to "number of degrees"

. Reply 3)

Revised.

  1. Line 49, please consider changing "finding out" to "determining"

Reply 4)

Revised.

  1. Line 49, please consider changing "athlete-specified" to "athlete-specific"

Reply 5)

Revised.

  1. Line 58, please replace "becomes" with "became"

Reply 6)

Revised.

  1. Line 63, [16] instead of [1616]?

Reply 7)

Revised.

  1. Line 135, please check the value of the Aref, the one in the text seems to be a little to large given the size of the ski defined a few lines earlier.

Reply 8)

The reference area Aref is defined as the area of the upper surface of the ski, consisting of three components, i.e., the rectangular region, the semi-circle ski head, and the up-tilting ramp, as follows,

  1. Line 157, please give unit of the kinematic viscosity.

Reply 9)

Revised.

  1. Lines 210-214, please provide a more detailed explanation of how CL and CD are related to Eqs. (9) and (10).

Reply 10

The description of Eqs. (9) and (10) have been revised to explicitly relate lift and drag with their dimensionless coefficients.

  1. Lines 434-443, are the references [6], [7], and [8] appropriate? Do they refer to ski jumping?

Reply 11)

We are sorry for the confusion led by these references. The references [6] and [8] (now Ref. [6] and [7] in the revised manuscript) are related to alpine skiing, in which aerodynamic forces also play an important role. Therefore, we retain these two references, and a brief explanation is included in the introduction. The reference [7] in the original manuscript has been removed.

  1. Lines 459-460, Forschungsinstitut

Reply 12)

Revised.

  1. Figures 6, 7, 8, pressure and its unit, p/Pa, would be better reported as pressure [Pa] or pressure (Pa)

Reply 13)

Revised.